# The Thyroid Hormone and Immunoglobulin Concentrations in Blood Serum and Thyroid Gland Morphology in Young Hens Fed with Different Diets, Sources, and Levels of Iodine Supply

**DOI:** 10.3390/ani13010158

**Published:** 2022-12-31

**Authors:** Maja Słupczyńska, Dorota Jamroz, Janusz Orda, Andrzej Wiliczkiewicz, Piotr Kuropka, Barbara Król

**Affiliations:** 1Department of Animal Nutrition and Feed Science, Wrocław University of Environmental and Life Sciences, 50-375 Wrocław, Poland; 2Department of Biostructure and Animal Physiology, Wrocław University of Environmental and Life Sciences, 50-375 Wrocław, Poland

**Keywords:** iodine, laying hens, thyroid hormones, thyroid gland morphometric, immunoglobulins

## Abstract

**Simple Summary:**

Bioactive micronutrients, such as iodine, regulate many key metabolic pathways in the body. Iodine is responsible, above all, for proper functioning of the thyroid gland and the hormones it secretes, which, in turn, determine the correct course of many processes. There is a close relationship between the level and source of iodine in the diets of animals and the activity of the thyroid gland. In addition, attention should also be paid to the presence in the diet of compounds that can decrease iodine utilization, such as the goitrogenic substances present in some feed materials. The examined effects of various sources and levels of iodine, as well as the presence of rapeseed meal in laying hen diets, indicated that the applied iodine sources and levels had no influence on thyroid hormone and immunoglobulin concentrations. However, iodine presence in the rapeseed meal diet and higher iodine concentrations can increase free triiodothyronine concentrations in blood serum. The level and source of iodine had an effect on the follicular diameter and height of the follicular epithelium cells.

**Abstract:**

The aim of this study was to examine the influence of the level (1, 3, and 5 mg I/kg) and source of iodine (KI, Ca(IO_3_)_2_, and KIO_3_) on thyroid hormone and immunoglobulin concentrations in the blood serum of laying hens alongside a histological picture of the thyroid. In the first, birds were fed grain–soybean meal mixtures, and in the second, two kinds of diets based on corn–soybean or corn–soybean–rapeseed meal were applied. In the experiments, we determined the levels of the blood serum thyroid hormones fT_3_ and fT_4_, as well as the morphological structure of the thyroid gland. In the second experiment, the concentration of immunoglobulins in blood serum was assayed. In both experiments, no influence of iodine source on thyroid hormone concentration was observed. However, increasing the iodine level in the full mixture and adding rapeseed meal in both experiments caused an increase in fT_3_ concentration. Increasing I-addition in both experiments led to a decrease in thyroid gland follicle diameter. Rapeseed meal inclusion (at a level of 10%) to the complete hen mixture led to an increase in thyroid gland follicle diameter. Applying KIO_3_ as an iodine source in both experiments caused a decrease in the thyroid gland height of follicle epithelial cells. Immunoglobulin concentrations in the serum were not affected by experimental factors. The results suggest that the methodologies of studies on the bioavailability of minerals and the corresponding analytical methods require unification. The lack of such standardization makes it impossible to engage in a satisfactory discussion of the results and exchange experiences.

## 1. Introduction

The thyroid gland has multiple functions and activities, and the dynamics of triiodothyronine (T_3_) and thyroxine (T_4_) creation depend on the supply of proper amounts of essential iodine in feeds. Iodine-containing hormones are involved in the growth and development of organisms and are important for the productivity of animals. Avian thyroglobulins are highly iodized molecules (1.5% of I) and can contain about 50–90 atoms of iodine [1]. The majority of thyroid hormones are secreted as T_4_ but can be reverted back into reversible T_3_ (rT_3_) via deiodination [2]. The T_3_ hormone was observed to be more metabolically active than T_4_ [3,4]. However, Shellabarger [3] confirmed that in birds, T_3_ and T_4_ are similarly effective. There is a close relationship between iodine levels in a diet and thyroid gland activity, which, in turn, regulates essential metabolic processes in the organism, especially brain function; heat production; thermoregulation; energy and lipid metabolism, including cholesterol, cellular oxidation processes, and protein synthesis; and cell and tissue maturity [5,6,7,8,9,10,11]. Moreover, numerous investigations found significant impacts of the thyroid on blood circulation, nervous system functions, reproduction, and skin regeneration [12,13,14]. Thyroid gland iodine-dependent hormones are responsible for metabolic processes that play an important role in immunoresponse [1,15,16]. The content of iodine in diet and its binding with thyrosine regulates the complex conversion of different thyronine forms and thyroxine [17,18,19]. The biosynthesis of thyroid hormones in an organism encounter the binding of iodine mainly through potassium iodine and the accumulation of I during the iodinization of tyrosine and the formation of triiodothyronine (T_3_) and thyroxine (T_4_). Large doses of iodine and their application in an organic form inhibit the conversion of thyroxine (T_4_) and can evoke a transmitted secretion of T_4_ from hepatocytes into blood serum [2,17,20]. The concentration of T_3_ in serum also depends on dietary carbohydrate levels [21] and other factors. A high content of carbohydrates in the diet is connected with an increased concentration of serum T_3_. The most important proteins for the creation of a matrix for thyroid hormone biosynthesis are thyreoglobulins [22]. The mechanisms of iodine management in the organism and thyroid gland are very complex. The turnover time of iodinated proteins synthetized continually in the thyroid is strictly connected with I-supply. A deficiency in I-supply leads to increased iodination of slowly metabolized unusual proteins, which can be used in the recycling process in the organism [13].

The supplementation of an animal’s diet with iodine is a common practice. Iodine in poultry is used mainly in the form of inorganic salts such as potassium iodide (KI), potassium iodate (KIO_3_), and calcium iodate (Ca(IO_3_)_2_), which are added into the mineral mixtures or premixes [23,24,25]. Numerous studies have shown that the bioavailability of iodine from these chemical forms can vary [25,26,27,28,29], which should also be taken into account when establishing recommended levels of iodine for the nutrition of farm animals, including poultry. According to Polish recommendations for laying hens, iodine levels in the complete mixture should be 1 mg per 1 kg [30]. According to EU Commission regulations [31], the upper tolerated level of dietary I recommended for laying hens is 5 mg/kg. Goitrogenic substances, an iodine transfer antagonist, present in feeds originating from the *Cruciferae* family, can act similarly to thiouracil inhibitors of triiodothyronine (T_3_) synthesis in thyroid tissue and may reduce the profitable activities of iodine supply in the diet [32,33,34]. Moreover, a long-term excess of iodine in rations (200–500 mg I/kg) can produce dangerous toxicosis in organisms and disorders in the conversion of triiodothyronine to thyroxin [9,17,18,19]. Taking the above into consideration when using iodine additives, particular attention should be paid to the form, level, and presence of antagonistic substances in the diets of animals because these factors may have an effect on the animals’ iodine supply.

Under this background, the aim of the present study was to examine the effects of various sources and levels of iodine, as well as the presence of rapeseed meal, in laying hen diets on thyroid hormones and immunoglobulin concentrations. We also present a thyroid histological picture. This paper presents data derived from two individual experiments that were conducted with the same line of lying hens under similar environmental conditions.

## 2. Materials and Methods

### 2.1. Animals and Management 

Experiment I (Exp. I) was carried out in the winter–spring and early summer periods, while experiment two (Exp. II) was carried out in the summer–autumn period. The first experiment lasted 180 days, and the second lasted 150 days. In both experiments, sixteen-week-old Hy-Line variety brown pullets were placed into battery cages and, before the laying period, were fed with a standard mixture containing about 145 g/kg of crude protein and 11.5 MJ of metabolizable energy. After attaining about a 30% laying rate, at the age of about 18 weeks, the 216 hens were allocated into cages according to body weight and laying rate at the starting time. In Exp. I, the cages were randomly divided into six treatments, each involving twelve replications, with thirty-six hens per treatment. In Exp. II, the cages were randomly divided into twelve treatments, each involving six replications with three hens per cage, for a total eighteen hens per treatment (Table 1). 

In both experiments, free access to water (nipples) was ensured, and the environmental conditions were registered. The room temperature varied between 16 and 25 °C, and the lighting program was set up for fourteen hours of light daily. Management was performed in compliance with European Union regulations, and The Local Ethics Commission for Experiments with Animals accepted all experimental procedures. The experiments were carried out in an experimental henhouse belonging to the Wrocław University of Life Sciences in Pruszowice. 

### 2.2. Diets and Feeding Program

The diet composition in both experiments was based on our own chemical analyses of feed components and calculated using linear optimization. The energy concentration was calculated using the values given in European Tables [35]. In the first experiment, hens received diets based on maize, barley, and wheat with soybean meal. To the basal diet, we added 1 or 5 mg/kg iodine in the form of KI, Ca(IO_3_)_2,_ or KIO_3_ (Table 2). 

In the second experiment, two kind of diets were applied: one based on corn–soybean (CS) and the second based on corn–soybean–rapeseed meal (CSR). Each diet was enriched with iodine using either KI or KIO_3_ in amounts appropriate to the anticipated iodine supplement at levels of 1, 3, and 5 mg of I/kg of feed (Table 2). Experimental concentrate mixtures in each experiment were isoproteic and isoenergetic and were prepared prior to the experiment. A homogenous mixture of iodine within the feeds was obtained by preparing premixtures of iodine supplements with iodine-free mineral–vitamin premixes that were made for the purposes of the experiments.

### 2.3. Analytical Methods in Feed Mixtures

The chemical composition of the feed components and complete mixtures were determined according to standard methods [36]. The nitrogen content was determined using a Kjeltec 2300 Foss Tecator apparatus (Höganäs, Sweden), and crude protein was calculated by multiplying the nitrogen content by 6.25 (AOAC, Rockville, Maryland, 984.13). Crude fiber was assayed via the Henneberg–Stohmann method using an Fibertec Tecator (Höganäs, Sweden) apparatus (AOAC, 978.10). After prior wet mineralization of the samples with nitric acid (HNO_3_) using a MarsX apparatus (CEM Corporation, Matthews, NC, USA), calcium and sodium content in the feeds was determined through atomic absorption spectrophotometry using an AA 240 FS apparatus with SIPS 20 (Varian, Mulgrave, Australia, AOAC, 968.08). Phosphorus content was analyzed after previous wet mineralization with nitric acid (HNO_3_) and perchloric acid (HClO_4_) according to the ammonium vanadomolybdate method using a Specol 11 spectrophotometer (Carl Zeiss, Jena, Germany) at a wave length of 470 nm (AOAC, 965.17). 

To determine the amino acid composition, samples of feeds were hydrolyzed with 6 M hydrochloric acid (HCl) for 24 h at 110 °C. Then, amino acids were separated according to the Moore and Stein method [37]. To determine sulfur amino acids, the feed samples were oxidized (0 °C, 16 h) with formic acid and hydrogen peroxide (9:1/*v*:*v*) prior to HCl hydrolysis and then separated using an AAA 400 Ingos Analyzer (Prague, Czech Republic). The iodine content in diets was analyzed using a spectrophotometric method based on the Sandell–Kolthoff reaction [38] on the basis of the speed of reaction: 2 Ce^4+^ + As^3+^ → 2 Ce^3+^ + As^5+^ [39,40]. The principle of the assessment was to reduce Ce^4+^ to Ce^3+^ in the presence of As^3+^, where iodide ions acts as catalyst for this reaction. The analyses were carried out at the Department of Animal Nutrition and Feed Science of Wrocław University of Environmental and Life Sciences.

### 2.4. Blood Collection and Analysis

At the end of the experiments (on the 180th day of Exp. I, the age of laying hens was 42 weeks, and on the 150th day of Exp. II, the age of laying hens was 38 weeks), six hens from each treatment were randomly selected for blood sampling. Blood samples (5 mL) were taken into clean sterilised tubes from the vena brachialis by a specialized veterinarian. The serum was obtained via centrifugation of the blood at 1500× *g* for 15 min at room temperature and stored at 20 °C until analysis. The concentration of immunoglobulins (Ig) in the blood serum was determined via ELISA kits f. Bethyl Laboratories, Inc. The absorption of light across the solution—effects of a colorimetric reaction with the chromogenic substrate TMB (3, 3′, 5, 5′ tetramytylobenzidyne)—was measured using a Thermo Scientific Multiskan EX. The content of thyroid hormones free triiodothyronine (fT_3_) and free thyroxine (fT_4_) in the blood serum was determined using a chemiluminescence immunoassay (CLIA) with an Immulite 1000 apparatus and chemotest f. Siemens. Biochemical assays were then performed in an accredited analytical laboratory.

### 2.5. Histological Examinations

At the end of each experiment, six hens from each treatment were chosen. After killing the hens via cervical dislocation, the thyroid glands were immediately removed. The thyroid glands were stabilized and fixed in 4% aqueous buffered formalin solution with calcium carbonate in a ratio of 9:1 (*v*/*v*) for 72 h. Then, the thyroid samples were rinsed for 24 h under running water, dehydrated with an increased alcohol series, and embedded in paraffin. This biological material was cut with a microtome into 7 µm thick slides and stained with hematoxylin according to Delafield’s method, as well as with eosin using the A2A-NOVUM method according to Geidies [41]. The histological preparations were examined using an optical microscope Nicon Eclipse 801 in transmitted light. For each sample, about 15–20 pictures were reviewed. The microphotographs were obtained using a Canon PS66 Camera. Morphometric measurements were made using the Nis-elements AR software, and the follicle diameter and height of the epithelial cells of follicles were examined. The assays were carried out at the Department of Biostructure and Animal Physiology of Wrocław University of Environmental and Life Sciences.

### 2.6. Statistical Evaluation of Results

All numerical data were evaluated using one, two, or three factorial ANOVA in the Statistica® version 10 computer software [42]. The differences between treatments and experimental factors were tested according to the following statistical models:

yij = µ + αi + eij (for differences between treatments)

yjik = µ + αi+ βj + (αβ)ij + eijk (for kind of diet or inclusion level of I-supplement or their level), or 

yijkl = µ + αi+ βj +γk + (αβ)ij + (αγ)jk + (αβγ)ijk + eijkl

Where yij, yijk, and yjikl represent the variance associated with parameter α; µ is the overall mean; αi is the treatment effect; βj is the kind of diet, I-supplement, or their levels; (αβ)ij is the the interaction effect; and eij or eijk ( eijkl) is an error term.

The individual measurements were treated as experimental units, and differences between treatments and averages for experimental factors (means) were analyzed for significance (*p* < 0.05 or *p* < 0.01) using Tukey’s test. All data are presented as average values and are accompanied by the corresponding SEM (standard error of mean) values.

## 3. Results

### 3.1. Thyroid Hormone Concentration in Hen Blood Serum

In the first experiment, the concentration of total free triiodothyronine (fT_3_) and free thyroxine (fT_4_) in individual treatments was similar, with no differences between treatments. However, there was a significant (*p* = 0.020) influence of iodine level addition into the diet on the fT_3_ concentration in blood serum. A larger iodine addition caused an increase in fT_3_ concentration from 4.94 to 5.83 pmol/L (Table 3). The types of I-bonds, KI, Ca(IO_3_)_2,_ or KIO_3,_ did not significantly influence thyroid hormone concentration in the serum.

In the second experiment, the content of both thyroid gland hormones in the blood serum was insignificantly diversified via experimental treatments. However, a comparison of the results concerning fT_3_ concentration within each dietary treatment showed that increasing iodine content with the same source in the diet from 1 to 5 ppm caused an increase in fT_3_ concentration but a decrease in fT_4_ concentration. The kind of diet, iodine source, and level of iodine supplement had no influence on the fT_4_ concentration in hen blood serum. On the other hand, the addition of 10% rapeseed meal to the hens’ diet significantly enhanced (*p* = 0.006) fT_3_ concentration, from 8.71 in the CS diet to 11.24 in the CSR diet (mean; Table 3). Similarly, the highest dietary I-supplementation content at 5 mg/kg caused an increase in fT_3_ content to 11.28 pmol/L in comparison to the fT_3_ concentration observed in the hen blood serum after 1 or 3 mg of I-supply to the diets (a mean of 9.3 pmol/L fT_3_). 

### 3.2. Thyroid Gland Structure

The analyzed material for the thyroid gland structure of hens in experiment I (Figure 1) presented a normal histological picture. These glands, which were derived from birds in all treatment groups, did not deviate from the healthy organs and were typical for the time of year in which the experiment was conducted.

In the group of birds fed the I-additives in an amount of 1 mg/kg, thyroid follicles were statistically (*p* = 0.000) larger (average above 70 μm), entirely filled with colloid, and surrounded by a dense network of blood vessels (Table 4). Follicular epithelial cells took the form of a cube and were slightly flattened in places. The thyroid cell surface was irregular, which may indicate a large mobilization of both secretory processes and colloid resorption. In the group of birds fed a higher dose of iodine, the thyroid follicles were about ¼ smaller (about 50 μm), whereas the epithelial cells were slightly larger. This result demonstrates the positive effects of additives on the process of thyroid hormone secretion.

The source of iodine applied in the experiment had no significant influence on the follicle diameter of the thyroid gland and ranged from 62.9 for KI to 67.1 for KIO_3_. The opposite relationship was noted for the height of the follicle epithelium cells. The iodine source significantly (*p* = 0.001) affected this parameter. The height of follicle epithelial cells was the highest for Ca(IO_3_)_2_ and the lowest for KIO_3_ (4.88 vs. 3.26 µm, respectively). 

In experiment II, the analyzed thyroid gland structure of the hens presented a normal histological picture (Figure 2). In the microscopic picture, we observed a small amount of connective tissue filling the space between the follicle-containing colloid. The shapes of follicular epithelial cells resemble the squamous epithelium, with some visible areas showing active resorption of colloid where the epithelium becomes larger and resembles a cuboidal epithelium. 

Morphometric analysis of the thyroid gland structure showed significant differences in particular treatments (Table 4). However, due to the existing interactions, these results are difficult to interpret. We observed that the main experimental factors (kind of diet, iodine source, and level) had an influence on both the follicle diameter and the height of follicle epithelial cells. Follicle diameter was larger (*p* = 0.011) in animals fed with a diet containing rapeseed meal (76.2 µm), but this factor did not have an influence on follicle epithelial cell height. The source of iodine used in experiment II did not change the thyroid gland follicle diameter. However, this parameter was reported to have an effect on the height of follicle epithelial cells. This effect was stronger in the thyroid glands of hens that received KI as a source of iodine (4.04 µm). Increasing iodine levels in the diets clearly decreased (*p* = 0.005) follicle size from 83.1 (for 1 mg of I/kg) to 67.8 µm for 5 mg of I/kg. A reverse relationship was noted for the height of follicle epithelial cells. The highest value for this parameter (4.24 µm) was found in hens receiving iodine addition of 5 mg/kg and the lowest in those receiving 1 mg/kg of iodine.

### 3.3. Immunoglobulin Concentration

The types of diets, iodine sources, and levels of I-supply did not produce a significant influence on the immunoglobulin concentration in blood serum (Table 5).

The mean level of IgA was 0.337 g/L, that of IgM was 1.168 g/L, and that of IgG was 11.33 g/L. The IgM and IgA concentrations were near the levels determined in healthy hens, but the IgG amounts in blood serum were relatively high.

## 4. Discussion

Thyroid hormones in blood plasma demonstrate great variability and can vary depending on animal age or season [5,33]. An important factor that can affect the values of obtained results concerning thyroid hormone concentration is the applied analytical method [24,43]. The use of two methods to analyze thyroid hormones in the serum of broiler breeders [44] (the chemiluminescence immunoassay (CLIA) or enzyme-linked immunosorbent assay (ELISA) methods) could diversify the observed concentrations of examined hormones, which could create difficulties in interpreting the obtained results. The study in [44] revealed the following values using CLIA and ELISA, respectively: T_4_: 31.6 and 21.3 nmol/L; T_3_: 1.47 and 1.08 nmol/L; fT_4_: 10.8 and 7.2 pmol/L; and fT_3_: 4.6 and 6.3 pmol/L. Stojević et al. [45] noted that in 28-day-old chickens, the triiodothyronine (T_3_) level could range between 1.2 and 2.0 nmol/L; at the age of 35 days, between 1.3 and 2.3 nmol/L; and at the age of 42 days, between 1.8 and 3.9 nmol/L, while the thyroxine (T_4_) concentration increased from 16–38 to 18–42 and 24–38 nmol/L, respectively. Together with aging, the plasma hormone levels can vary between 3.6 and 1.4 T_3_ ng/mL and 6.6 and 18.4 T_4_ ng/mL [46]. The concentrations of thyroid hormones in the blood serum of chickens given by Ma et al. [47] were 1.3–1.7 nmol/L for T_3_, 39–49 nmol/L for T_4_, 1.7–3.14 pmol/L for fT_3,_ and 6.1–8.0 pmol/L for fT_4_. Similarly, the age of adult laying hens (22-34 weeks) has a significant impact on thyroid hormone levels in plasma, which were 0.97–2.93 nmol/L for T_3_, 25–39 nmol for T_4_, 10.0–14.5 pmol/L for fT_4,_ and 3.8–18.4 pmol/L for fT_3_ [48]. 

The results for the concentration of thyroid hormones obtained in both experiments were within the values given by the above-mentioned authors. This result may indicate that the applied experimental factors did not cause dysregulation in the thyroid hormone balance. However, the seasonal fluctuations in hormone concentration in blood serum cannot be clearly stated, while the hormones were determined in two different hen groups. 

Available literature data indicate that the applied iodine levels influenced the concentration determined in the blood serum thyroid hormone concentration. The iodine supplementation of diets caused an insignificant decrease in the T_3_ concentration (1.98–2.33 nmol/L) in pig blood plasma and a significantly (*p* < 0.01) increased level of T_4_ from 50.4 to 95.9 nmol/L [49]. By adding 10 µg iodine in the form of KIO_3_ to 1 mL of drinking water for rats, T_4_ increased to 16.3 µg/dL compared to the control animal (10 µg/mL), fT_4_ increased to 5.74 (control: 2.83 pmol/L), and fT_3_ increased to 2.31 pmol/L (control: 2.12 pmol/L) [14]. Röttger et al. [25] observed significant growth of the iodine concentration in the thyroid gland due to I-supply in the diet using 4 or 10 mg I/kg of feed in the form of KI or KIO_3_. Li et al. [11] found no impact on the T_3_ thyroid gland, but other authors noted a clear response of thyroid activity to high I-levels in the diet [50] or increased thyroid stimulating hormone pituitary TSH concentrations. For experiments with broilers, Behroozlak et al. [51] observed that the dietary inclusion levels of iodine supplementation (0, 2.5, and 5 mg/kg) as KI had no influence on the T3 concentration, unlike the T4 concentration, which significantly increased from 12.45 (with an iodine concentration of 2.5 mg/kg) to 14.11 nmol/L (with iodine concentration 5 mg/kg). In experiments with laying hens fed over a period 12 weeks with a diet containing various levels of CIOD and EDDI as supplemental iodine sources (2.0, 4.0, and 8.0 mg/kg of feed), Sarlak et al. [24] observed that both experimental factors had an effect on serum T3 and T4 concentrations. In hens that received EDDI as an I-supplement after 12 weeks of experimentation, both T3 and T4 concentrations increased in comparison with the animals that received CIOD (T3 and T4 concentrations for EDDI were 1.97 and 16.02 ng/mL, respectively, whereas these values for CIOD were 1.81 and 15.47 ng/mL, respectively). In our experiments, we observed no effect of iodine source on fT3 and fT4 concentrations. In a previous experiment [24], only the highest concentration of iodine (8.0 mg/kg of diet) led to an increase in T3 and T4 concentration to 2.75 and 17.78 ng/mL, respectively. This result is in agreement with the data from our own studies with respect to fT_3_ concentration: increasing the level of I-supplementation led to an increased fT_3_ concentration in both experiments. Contrary to the cited data, no effect of iodine levels on the fT4 concentration was observed. In an experiment by Ahmadi M. [52], different levels of rapeseed meal addition (5%, 10%, and 15%) to the diet had no significant effect on T3 and T4 concentrations. On the other hand, Maroufyan E. and Kermanshahi H. [53], in experiments with broilers, noted that increasing the rapeseed level from 7.5% to 15% in the full mixture led to an increase of serum T3 concentration on the 42nd day of the experiment (1.50 vs. 1.90 ng/mL, respectively), while there was no effect on T4 concentration. Data from our experiment (Exp. II) showed a similar effect of rapeseed meal addition on fT_3_ and fT_4_ concentrations. 

The small seasonally variable morphology of the thyroid gland in birds has been observed by other authors. In winter, due to the shorter amount of daylight, thyrotropin (TSH) and thyroid hormone activities are reduced, leading to greater heat production and thermoregulation; consequently, the follicles are larger than 100 µm, and the follicular secretion cells have a flattened shape to a height of about 2 µm. In the summer, the follicles have a diameter of about 60 µm, and the height of the follicular epithelial cells reaches up to 12 µm [54,55,56,57]. Additionally, age-specific changes can be observed in birds, where the follicles become larger (average), and the height of the epithelial cells decreases. Our studies of follicular size measurements did not provide clear answer regarding the effects of dietary I-supplementation. The numerous significant differences between treatments in Exp. I and II, as well as the significant interactions between experimental factors, made interpretation of the results very difficult. Factorial analysis showed a significant decrease in follicular size based on the I-level in the diet (Exp. I), without influence from the iodine source. An inverse relationship between the level and source of iodine was found for follicle epithelial cell height. In experiment II, the use of rapeseed meal in the diet significantly enlarged the follicle size; by supplying 5 mg I/kg, a decrease in follicle diameter was observed. Using KIO_3_ as an I-source decreased the height epithelial cells compared to the effects of KI. The highest epithelium was observed after using 5 mg I/kg I-addition.

A greater follicle size indicates the accelerated mobilization of colloid, but in the case of the analyzed material, this mobilization seems to yield slower growth of the follicles. In the case of treatments with the addition of rapeseed meal, there are fewer follicular cells, which may indicate a lower potential for future growth (i.e., fewer cells that can enter the follicle). The relationship between follicle size and their number present in the area of the microscopic picture adjacent to the bladder offers potential for hypertrophy, as well as synthesis rate and colloid mobilization. Thyroid hormones have two stages: the original cells and cells separated into the light bladder before resorption, at which point the condenser must be enriched with iodine in colloid. Therefore, the synthesis of thyroid stimulating hormone is associated with the accelerated synthesis of colloid and its resorption. In the initial phase, this phenomenon means an increase in the synthesis of colloidal and larger bubbles and their subsequent resorption. In animals, this stage is followed by stabilization of the vesicles, and these processes adopt all classical structures. An increase in the volume of thyroid follicles can thus be caused by colloid synthesis acceleration and deceleration of its resorption. These phenomena yield a slowdown in metabolism in autumn and winter but rapid acceleration in spring and summer.

In the present study, (Exp. II), we did not observe any influence of the applied diets and iodine sources or levels on serum immunoglobulin concentrations. Mehner and Hartfiel [43] reported that the individual variability of IgG usually amounts to 5.5–8.2 g/L but can oscillate between 2.79 and 14.45 g/L with variability of about 564%. In quails, Niemiec [58] observed very low concentrations with 16–21.9 µg/dL IgM, 4.9–6.4 µg/dL IgG, and 3.8–4.7 µg/dL IgA. The long-term fortification of iodine content in the hens’ diet had no negative influence on the physiological and biochemical parameters in the blood or the good health of the birds [29]. Additionally, thyroid gland morphology and immunoglobulin concentrations in the blood serum corresponded to correct thyroid functions in laying hens.

## 5. Conclusions

The experimental factors did not show any negative effects on changes in the structure and functioning of the thyroid gland, which may indicate the possibility to safely use iodine supplementation in laying hens over a long period of time, even in higher amounts. Observed changes in fT_3_ and fT_4_ concentrations as an effect of increased iodine levels in the diet may indicate a shift in the balance towards a more metabolically active form of the hormone. There was no effect of diet, level of iodine, or I-supplementation on immunoglobulin concentrations in hen blood serum. 

Overall, the methodologies of studies on the bioavailability of minerals and the corresponding analytical methods require unification. The lack of such standardization makes it impossible to develop a satisfactory discussion of the results and exchange experiences. Another problem in interpreting the data is the number of factors that are simultaneously taken into account in such experiments.

## Figures and Tables

**Figure 1 animals-13-00158-f001:**
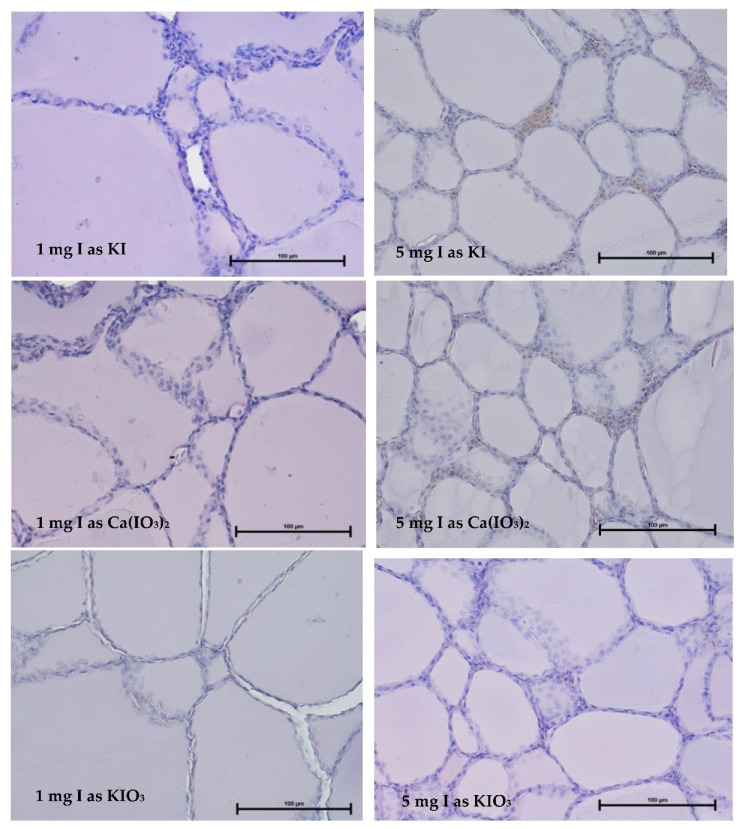
Representative pictures of the thyroid gland from the birds fed with diets of different iodine supplements and levels (HiE Mag 400) (Experiment I).

**Figure 2 animals-13-00158-f002:**
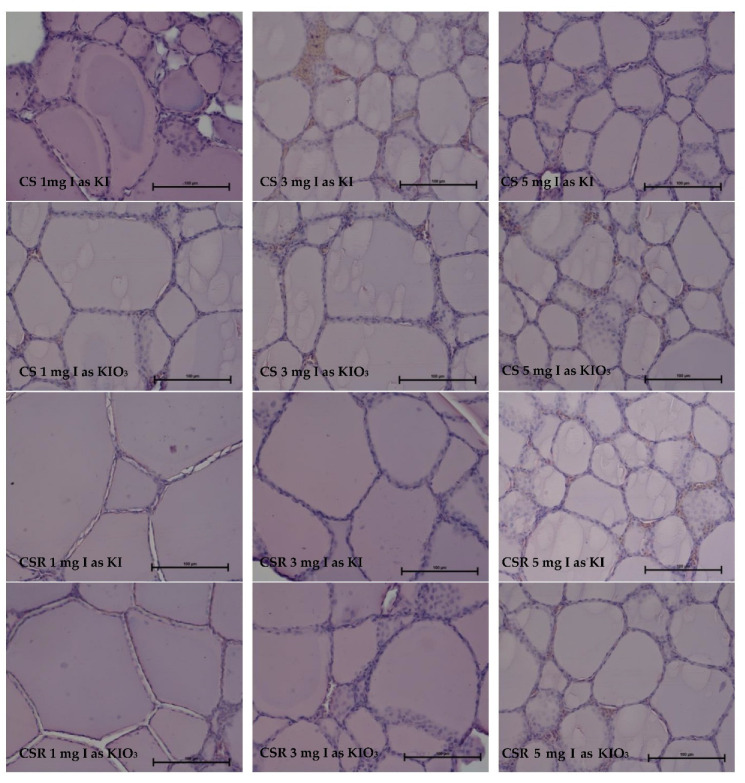
Representative pictures of the thyroid gland from the birds fed different diets, with different iodine supplements and levels (HiE Mag 400) (Experiment II).

**Table 1 animals-13-00158-t001:** Design of the experiments.

Experiment I											
Type of iodine supplement											
potassium iodide KI				calcium iodate Ca(IO_3_)_2_				potassium iodate KIO_3_			
Level of I-supplement mg/kg											
1		5		1		5		1		5	
**Experiment II**											
Treatments—Diets											
CS *						CSR **					
Type of iodine supplement											
potassium iodide KI			potassium iodate KIO_3_			potassium iodide KI			potassium iodate KIO_3_		
Level of I-supplement mg/kg											
1	3	5	1	3	5	1	3	5	1	3	5

* corn–soybean-based mixture; ** corn–soybean–rapeseed-based mixture.

**Table 2 animals-13-00158-t002:** Basal composition of the experimental diets in both experiments.

Ingredient, % of Diet	Diets		
	Experiment I	Experiment II	
		CS	CSR
Corn meal	15.00	15.00	15.00
Wheat meal	25.81	28.44	24.09
Barley meal	20.00	20.00	20.00
Soybean meal solvent extracted	21.50	20.18	13.23
Rapeseed 00 meal. solvent extracted	-	-	10.00
Rapeseed oil	5.20	4.28	5.71
Dicalcium phosphate	2.05	0.481	0.435
Chalk	7.25	8.931	8.851
NaCl	0.37	0.015	0.013
DL methionine	0.184	0.046	0.024
L-Lysine	0.135	0.128	0.148
Vitamin and mineral premix ^1^	2.50	2.50	2.50
Nutritive value of mixtures			
EM, MJ/kg	11.65	11.65	11.65
Crude protein, g/kg	163.50	165.00	162.00
Crude fiber, g/kg	33.00	36.26	28.97
Ca, g/kg	38.80	38.80	38.90
P available, g/kg	4.30	4.30	4.32
Na, g/kg	1.75	1.70	1.71
Methionine, g/kg	4.32	4.30	4.31
Lysine, g/kg	9.08	9.00	8.97
Nutritive recommendations for Hy-Line laying hens			
EM, MJ/kg	11.65		
Crude protein, g/kg	165.00		
Crude fiber, g/kg	to 40.00		
Ca, g/kg	38.80		
P available, g/kg	4.30		
Na, g/kg	1.70		
Methionine, g/kg	4.30		
Lysine, g/kg	9.00		

^1^ Supplied per kilogram of diet: vitamin A—12,000 IU; vitamin D_3_—3300 IU; vitamin E—30.0 mg; vitamin K_3_—3.0 mg; vitamin B_1_—2.20 mg; vitamin B_2_—6.60 mg; vitamin B_6_—3 mg; vitamin B_12_—27 mcg; pantothenic acid—10.2 mg; nicotinic acid—30 mg; folic acid—0.9 mg; choline—0.315 mg; Mg—113 mg; Zn—72 mg; Co—0.48 mg; Cu—10.0 mg; Fe—60 mg, Se—0.30 mg, I—0.0 mg.

**Table 3 animals-13-00158-t003:** Thyroid hormone concentration in hen blood serum (Exp. I and Exp. II).

Item	Experiment I		Experiment II	
	fT_3_, pmol/L	fT_4_, pmol/L	fT_3_, pmol/L	fT_4_, pmol/L
Treatments				
Experiment I				
1 mg I as KI	5.14	10.71		
5 mg I as KI	5.48	11.47		
1 mg I as Ca(IO_3_)_2_	5.05	22.69		
5 mg I as Ca(IO_3_)_2_	5.54	11.85		
1 mg I as KIO_3_	4.64	9.92		
5 mg I as KIO_3_	6.45	13.82		
Experiment II				
CS				
1 mg I as KI			8.51	11.06
3 mg I as KI			7.01	11.58
5 mg I as KI			10.14	9.52
1 mg I as KIO_3_			7.93	8.02
3 mg I as KIO_3_			9.04	11.36
5 mg I as KIO_3_			9.66	10.04
CSR				
1 mg I as KI			11.19	8.55
3 mg I as KI			9.29	7.22
5 mg I as KI			13.29	7.28
1 mg I as KIO_3_			10.92	10.32
3 mg I as KIO_3_			10.70	7.59
5 mg I as KIO_3_			12.02	7.77
SEM	0.189	2.149	0.444	0.600
*p* value	0.111	0.492	0.137	0.920
Kind of diet				
CS			8.71 ^A^	10.26
CSR			11.24 ^B^	8.12
Iodine source				
KI	5.30	10.80	9.90	9.20
Ca(IO_3_)_2_	5.29	20.13	-	-
KIO_3_	5.54	11.30	10.05	9.19
Level of iodine supplement				
1 mg	4.94 ^a^	15.62	9.64 ^Aa^	9.49
3 mg	-	-	9.01 ^Ab^	9.43
5 mg	5.83 ^b^	12.52	11.28 ^B^	8.65
Source of variation, *p* value				
Kind of diet	-	-	0.006	0.181
Iodine source	0.838	0.377	0.830	0.891
Iodine level	0.020	0.616	0.045	0.832
Kind of diet × I-source	-	-	0.874	0.576
Kind of diet × I-level	-	-	0.826	0.439
I-source × I-level	0.208	0.335	0.372	0.880
Kind of diet × I-source × I-level	-	-	0.744	0.769

Means within column marked using ^a^ and ^b^ superscripts differ significantly at *p* < 0.05, means within column marked using ^A^ and ^B^ superscripts differ significantly at *p* < 0.01.

**Table 4 animals-13-00158-t004:** Morphometric characteristic of the thyroid gland (Exp. I and Exp. II).

Item	Experiment I		Experiment II	
	Follicle Diameter, µm	Height of Follicle Epithelial Cells, µm	Follicle Diameter, µm	Height of Follicle Epithelial Cells, µm
Experiment I				
1 mg I as KI	71.6 ^A^	3.87		
5 mg I as KI	54.2 ^Bc^	4.21		
1 mg I as Ca(IO_3_)_2_	72.3 ^A^	4.58		
5 mg I as Ca(IO_3_)_2_	53.9 ^Bc^	5.17 ^a^		
1 mg I as KIO_3_	75.1 ^Ab^	3.22 ^b^		
5 mg I as KIO_3_	59.0 ^Bc^	3.31 ^b^		
Experiment II				
CS				
1 mg I as KI			72.5	3.87
3 mg I as KI			65.2 ^abce^	4.58 ^ABab^
5 mg I as KI			54.2 ^Ac^	4.21
1 mg I as KIO_3_			87.1 ^Bde^	3.22 ^BCbcd^
3 mg I as KIO_3_			76	2.59 ^Cd^
5 mg I as KIO_3_			59.1 ^ACDabc^	3.31 ^bcd^
CSR				
1 mg I as KI			88.7 ^Bd^	2.90 ^BC^
3 mg I as KI			81.3 ^BCDacde^	3.78 ^cd^
5 mg I as KI			67.5	5.17 ^Aa^
1 mg I as KIO_3_			84.1 ^BDde^	3.09 ^BCbcd^
3 mg I as KIO_3_			78.1 ^abde^	3.16 ^BCbcd^
5 mg I as KIO_3_			57.5 ^ACb^	4.43 ^abc^
SEM	1.640	0.166	1.534	0.111
*p* value	0.000	0.009	0.000	0.000
Kind of diet				
CS			69.0 ^a^	3.64
CSR			76.2 ^b^	3.76
Iodine source				
KI	62.9	4.04	71.6	4.04 ^A^
Ca(IO_3_)_2_	63.1	4.88 ^A^		
KIO_3_	67.1	3.26 ^B^	73.7	3.29 ^B^
Level of iodine supplement				
1 mg	72.9 ^A^	3.89	83.1 ^A^	3.43 ^A^
3 mg			75.2	3.51
5 mg	59.1 ^B^	4.23	67.8 ^B^	4.24 ^B^
Source of variation, *p* value				
Kind of diet			0.011	0.528
Iodine source	0.097	0.001	0.456	0.000
Iodine level	0.000	0.300	0.005	0.000
Kind of diet × I-source			0.004	0.049
Kind of diet × I-level			0.001	0.004
I-source × I-level	0.001	0.828	0.019	0.083
Kind of diet × I-source × I-level			0.000	0.487

Means within column marked with ^a, b, c, d^ and ^e^ superscripts differ significantly at *p* < 0.05, means within column marked with ^A, B, C^ and ^D^ superscripts differ significantly at *p* < 0.01.

**Table 5 animals-13-00158-t005:** Immunoglobulin concentration in hen blood serum (g/L) (Exp. II).

Item	IgA	IgM	IgG
Treatments			
CS			
1 mg I as KI	0.397	1.398	11.99
3 mg I as KI	0.284	1.103	10.22
5 mg I as KI	0.367	1.326	11.08
1 mg I as KIO_3_	0.280	1.236	11.81
3 mg I as KIO_3_	0.403	0.946	11.67
5 mg I as KIO_3_	0.284	0.985	11.57
CSR			
1 mg I as KI	0.285	1.255	10.32
3 mg I as KI	0.349	1.080	13.73
5 mg I as KI	0.275	1.432	9.71
1 mg I as KIO_3_	0.453	1.065	12.00
3 mg I as KIO_3_	0.301	1.055	9.99
5 mg I as KIO_3_	0.372	1.140	11.93
SEM	0.019	0.079	0.470
*p* value	0.643	0.927	0.914
Kind of diet			
CS	0.336	1.166	11.39
CSR	0.339	1.171	11.28
Iodine source			
KI	0.326	1.265	11.17
KIO_3_	0.349	1.071	11.49
Level of iodine supplement			
1 mg	0.354	1.238	11.53
3 mg	0.334	1.046	11.40
5 mg	0.324	1.221	11.07
Source of variation,			
*p* value			
Kind of diet	0.942	0.757	0.811
Iodine source	0.483	0.354	0.602
Iodine level	0.857	0.589	0.906
Kind of diet × I-source	0.171	0.675	0.96
Kind of diet × I-level	0.935	0.410	0.672
I-source × I-level	0.939	0.639	0.548
Kind of diet × I-source × I-level	0.066	0.887	0.254

All mean within columns are statistically insignificant.

## Data Availability

Not applicable.

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
