# Peer review of "The Thyroid Hormone and Immunoglobulin Concentrations in Blood Serum and Thyroid Gland Morphology in Young Hens Fed with Different Diets, Sources, and Levels of Iodine Supply"

_animals, 2022, doi:10.3390/ani13010158_

Round 1

Reviewer 1 Report

The manuscript presented for the review presents the original results of studies that can make an important contribution to the current state of the art regarding the presented topic. In the reviewer's opinion, the Introduction indicates the state of current knowledge. However, it seems that there is no clearly defined research hypothesis. The experimental design is appropriate to resolve the stated objectives of the study. The experimental techniques are appropriate to resolve the stated objectives of the study. The results are presented in an unbiased fashion, a clear, concise and complete manner. The discussion is relevant and adequate for full interpretation of results. I submit the final conclusion to the authors' consideration. Is this conclusion not too far-fetched? How long did the experiment last? Is it really long enough for all the possible effects of iodine supplementation to occur? No statement of the Institutional Review Board Statement.

Reviewer 2 Report

The aim of this study was study was to examine the effect of various sources and levels of bioactive mineral elements, i.e. iodine supplements, as well as the presence of rapeseed meal in laying hen diet on the thyroid hormones and immunoglobulins concentration in blood serum of laying hens as well as thyroid histological picture.  The authors mentioned that data derived from two individual experiments that were conducted with the same line lying hens and in similar environmental conditions.

The introduction; The introduction was carried out correctly and with the appropriate depth in structure.

The hypothesis was written in appropiately in the paper.

The authors reported in both experiments no influence of iodine source on thyroid hormones concentration was stated. Rapeseed meal addition into the diet and increase of iodine concentration caused significant increase of fT3 concentration in blood serum. In the first experiment the lowest follicular diameter and height of follicular epithelium cells were observed in group received KIO3.In the second experiment iodine source and level affected height of follicle epithelial cells as well. Immunoglobulins concentration in the serum were not affected by experimental factors.

Their conclusion  were the experimental factors did not show a negative effect on changes in the structure and functioning of the thyroid gland, which may indicate the possibility of safe use of iodine supplementation in laying hens over a longer period of time, even usingits higher amounts.

Discussion and conclusion Written clearly, concisely and precisely

General comments.

As I commented, it is a well-prepared and structured scientific writing; however, I am struck by the fact that they do not show data on productive parameters; in the sense that what would be the implication or implications at the metabolic level in the changes of thyroid hormones; as well as at the immunological level? They could wave a little on the subject

Line 24. l (1, 3, 5 mg I/kg) , Is the unit of measure correct?

Line 85. The genetic provider of the lineage is not mentioned.

The regulation of some animal care and use committee is not mentioned.

Line 85. The genetic provider of the lineage is not mentioned.

The regulation of some animal care and use committee is not mentioned.

Table 2.   L-lizyne. Lizyne  Check the spelling please

 could you rewrite the material and methods. Being more precise in the age and duration of the experiment

Reviewer 3 Report

The manuscript "The immunoglobulins and thyroid hormones concentrations in blood serum and thyroid gland morphology in young hens fed with different diets, sources, and levels of iodine source" is well written and on an important topic for the physiology of laying hens.

However, in my opinion, it has some limitations.

Major

1) Lack of a group of animals not supplemented with iodine (control group). Without this group, it is not possible to know whether the supplementation of 1 mg/kg of iodine has any effect on the parameters analyzed. This data is critical to determine the effects of iodine supplementation.

Minors

2) Abstract.

There is no conclusion.

3) Introduction

The authors must make clear which hypothesis(s) of the study. For example, what are the expected effects of using different sources of iodine? Why use concentrations of 1, 3, and 5 mg I/Kg? What is the hypothesis for the use of rapeseed?

4) Materials and Methods

Authors must justify the use of three different sources of iodine; the iodine and rapeseed concentrations used, and why they excluded Ca(IO3)2 from experiment II.

5) Results

Authors should review figures 1 and 2. The images are not compatible with the results described in tables I and II. Se ee Fig 1E, follicle diameter 53.9 um; Fig 1F, follicle diameter 59 µm.

6) Discussion

Authors should make a new discussion following the parameters described in the manuscript research section of the Instructions for authors. In the current form, there is a repetition of the results and there is no comparison with previous studies (see lines 312 to 359)

7) References

Authors should make an effort to cite up-to-date references. The most recent reference is from 2014; since then, several studies have been published.

Round 2

Reviewer 3 Report

Dear authors, Thanks for the replies. The manuscript in its current form presents the results obtained more adequately.